# Modeling of Respiratory Diseases Evolving with Fibrosis from Organoids Derived from Human Pluripotent Stem Cells

**DOI:** 10.3390/ijms24054413

**Published:** 2023-02-23

**Authors:** Irene Chamorro-Herrero, Alberto Zambrano

**Affiliations:** Laboratory of Biotechnology of Stem Cells and Organoids, Chronic Diseases Program, Instituto de Salud Carlos III, 28220 Madrid, Spain

**Keywords:** pluripotent stem cells, minilungs, disease modeling, fibrosis, myofibroblasts, idiopathic pulmonary fibrosis, IPF, cystic fibrosis, CF, chronic obstructive pulmonary disease, COPD, SARS-CoV-2, COVID-19

## Abstract

Respiratory disease is one of the leading causes of morbidity and mortality worldwide. There is no cure for most diseases, which are treated symptomatically. Hence, new strategies are required to deepen the understanding of the disease and development of therapeutic strategies. The advent of stem cell and organoid technology has enabled the development of human pluripotent stem cell lines and adequate differentiation protocols for developing both airways and lung organoids in different formats. These novel human-pluripotent-stem-cell-derived organoids have enabled relatively accurate disease modeling. Idiopathic pulmonary fibrosis is a fatal and debilitating disease that exhibits prototypical fibrotic features that may be, to some extent, extrapolated to other conditions. Thus, respiratory diseases such as cystic fibrosis, chronic obstructive pulmonary disease, or the one caused by SARS-CoV-2 may reflect some fibrotic aspects reminiscent of those present in idiopathic pulmonary fibrosis. Modeling of fibrosis of the airways and the lung is a real challenge due to the large number of epithelial cells involved and interaction with other cell types of mesenchymal origin. This review will focus on the status of respiratory disease modeling from human-pluripotent-stem-cell-derived organoids, which are being used to model several representative respiratory diseases, such as idiopathic pulmonary fibrosis, cystic fibrosis, chronic obstructive pulmonary disease, and COVID-19.

## 1. Introduction

The respiratory system is composed of two main compartments: the airways and the alveoli. The airways consist of the nasal cavity, trachea, bronchi, and bronchioles. The airways conduct airflow to the distal alveoli in which gas exchange between the exterior and the underlying vasculature takes place. Clearance of microbes and suspended particles in the air is another essential function of the respiratory tree conducted by cells of the upper airway epithelium. The luminal surfaces of the airways have lining of ciliated pseudostratified columnar epithelium and contain goblet, ciliated, and basal cells, which are stem cells or progenitors of all the cells of the respiratory epithelium. As the degree of branching within the airway tree continues, the epithelium gradually changes from pseudostratified to simple cuboidal and the predominant cells become non-ciliated cells. Club cells are secretory cells of the bronchiolar epithelium and contribute to production of non-mucinous secretory proteins to the extracellular lining fluid. The alveoli are composed of two main epithelial cells: alveolar type I cells (ATI cells), implicated in gas exchange, and alveolar type II cells (ATII cells), responsible for secretion of the surfactant that reduces surface tension and promotes generation of ATI cells (Figure 1). Recent studies have described new epithelial cell types, such as CFTR-rich pulmonary ionocyte and respiratory airway secretory (RAS) cells [1,2]. The pulmonary ionocyte appears to be a major source of CFTR activity in the airway epithelium, suggesting a role in luminal pH regulation that could be relevant for cystic fibrosis physiopathology. RAS cells are primarily located in respiratory bronchioles and can serve as a distal lung progenitor for ATII cells.

Respiratory diseases may be caused by genetic disorders, infections, tobacco smoke, particle inhalations, air pollution, etc., and affect one in five people. For instance, chronic obstructive pulmonary disease (COPD), lower respiratory infections, and airway and lung cancers are the third, fifth, and sixth largest causes of global death, respectively (World Health Organization—https://www.who.int/news-room/fact-sheets/detail/the-top-10-causes-of-death, accessed on 20 February 2023; [3]). During or secondarily to progress of many respiratory conditions, fibrosis of the airways or the lung may eventually evolve. The fibrotic process is part of a general process represented by so-called “wound healing” in response to either endogenous and or exogenous damage and occurs in all organs in four common phases: (i) initiation due to injury of the tissue, (ii) inflammation and activation of effector cells, (iii) synthesis of extracellular matrix (ECM) components, and (iv) deposition of ECM to restore organ failure (remodeling phase) [4]. This process relies on multiple epithelial–fibroblast interactions. After tissue damage, the epithelial cells release inflammatory mediators regarding entry of leukocytes (neutrophils, macrophages, and T-cells) that secrete profibrotic cytokines, such as IL-1β, TNF, IL-13, and TGFβ. The activated macrophages and neutrophils remove dead cells and activate resident fibroblasts and other sources. Fibroblasts proliferate and differentiate into myofibroblasts that secrete ECM components (hyaluronic acid, fibronectin, proteoglycans, and interstitial collagens) to promote wound healing and restoration. Myofibroblasts exhibit increased synthetic capacities and are responsible for contractility of scar tissue [4,5,6,7]. Progenitors of myofibroblasts include resident fibroblasts, fibrocytes, smooth muscle cells, pericytes, epithelial and endothelial cells undergoing epithelial- or endothelial-to-mesenchymal transitions, stromal cells, or hepatic stellate cells [8]. Under normal conditions, this process of “wound healing” has a beginning and an end but may be impaired under persistent tissue damage, pathological states, or ageing. In these cases, resolution of the damage involves fibrotic lesions with excessive ECM deposition and presence of senescent and inflammatory cells [4,9,10]. The fibrotic tissue loses its elasticity due to ECM deposition, contracted fibroblasts, and reduced vasculature. Irreversible destruction of the lung architecture may lead to organ malfunction, disruption of gas exchange, and death from respiratory failure. Fibrotic remodeling of the lung may take place in the lung parenchyma (alveolus and lung matrix, e.g., in idiopathic pulmonary fibrosis, IPF) and the airways (e.g., in asthma, chronic obstructive pulmonary disease COPD, cystic fibrosis (CF)). However, IPF may involve airway-specific pathogenesis, such as bronchiolization of the distal airspace with abnormal airway cell-types and honeycomb cystic terminal airway-like structures [11].

Causes of pulmonary fibrosis include environmental pollutants, some drugs, radiation, connective tissue diseases, or interstitial lung disease; however, in most cases, no clear cause is found [4,12,13,14,15]. Thus, fibrosis of varying degrees has been described in smokers and several lung diseases, such as (i) obstructive lung diseases (asthma, bronchiolitis, chronic obstructive pulmonary disease (COPD), lung transplantation); (ii) infectious and suppurative lung diseases (cystic fibrosis (CF), pneumonia, etc.); (iii) adult respiratory distress syndrome (ARDS) and lung edemas; (iv) diseases specific to infants (chronic lung disease of prematurity, surfactant protein-B deficiency, etc.); and (v) interstitial lung diseases (sarcoidosis, idiopathic pulmonary fibrosis (IPF), hypersensitivity pneumonitis, silicosis, asbestosis), etc. [16,17,18,19,20,21,22,23,24,25,26,27,28,29,30,31,32]. Modeling of such complex pathogenic contexts that may affect lung structure and involve epithelial–mesenchymal interactions is challenging. During the last decade, we have observed great advances in biology of stem cells and establishment of human-stem-cell-derived organoid cultures in 3D structures. These 3D cultures consist of aggregates of cells that self-organize in structures that mimic relatively well the structure and function of the native organ [33]. The two most common types of organoids differ regarding the stem cells from which they emerge: pluripotent stem cells (PSCs) and organ-specific adult stem cells (ASCs). PSCs have a broader differentiation spectrum than ASCs and offer the possibility of generating unlimited individual-specific organoids. PSCs include both embryonic stem cells (ESCs) and induced pluripotent stem cells (iPSCs) [34,35,36,37]. Development of organoids-based models containing cells of the airways and the alveoli have enabled understanding of genetic disorders, chronic, and infectious diseases. This review discusses the use of PSCs-derived lung organoids (referred to here as minilungs) to model pulmonary fibrosis in the context of various human respiratory conditions, such as IPF, CF, COPD, and COVID-19. We will also discuss recent advances and propose possible lines of research development to study and model lung diseases that might evolve with fibrosis of different texture and magnitude.

## 2. Lung Organoids from Human Pluripotent Stem Cells (hPSCs)

Development of 3D minilungs using biotechnology applied to pluripotent stem cells has contributed significantly to the understanding of respiratory diseases. To generate organoids, PSCs are subjected to a differentiation protocol that largely reflects the normal differentiation events that occur during embryonic development. Current protocols have been refined and improved in terms of manipulation of stem cells and efficiency of differentiation. In essence, these protocols depend on generation of definitive endoderm and generation of progenitors. The signaling pathways involved are those regulated by Activin A, TGFβ, WNT, NOTCH, and BMP4 [38,39,40,41,42,43,44,45,46,47]. Use of specific agonists or inhibitors allows turning the corresponding pathway on or off at certain times of the sequential protocol. Wnt3a, Wnt agonist CHI99021, and BMP4 inhibitor Noggin are commonly used to generate lung progenitors that are subsequently matured by addition of fibroblast growth factors FGF2, FGF7, and FGF10 and BMP4 [43,46,48,49,50,51,52]. Thus, PSCs are driven to produce epithelial cells from airways and alveoli. The differentiated cells can be displayed as a bidimensional array (2D minilung) or in 3D [50,51,53,54,55]. Gotoh et al. reported generation of 3D structures from anterior foregut endoderm containing cells from the bronchiolar epithelium and alveoli [56]. However, Chen et al. managed to create minilungs showing branching structures with proximal–distal compartmentalization consisting of cells from the airways and alveoli with some mesenchyme [53]. These organoids, named “Lung Buds Organoids” or LBOs, showed an expression profile corresponding to the second trimester of human development and were particularly enriched in ATII cells. One year later, we described generation of new minilungs structurally different from the lung buds, named “paddle-racquet lung organoids” or PRLOs [55]. These PRLOs present globose structures that enclose wider lumens than those expressed by LBOs, probably reflecting alveolar-like structures resembling the typical alveolospheres found after differentiation of ASCs from alveoli (Figure 2). Generation of such PRLOs was achieved via addition of dexamethasone to the maturation cocktail at certain times of differentiation. PSCs-derived minilungs, in 2D or 3D formats, have allowed, for instance, modeling of infection of human respiratory syncytial virus (HRSV) [53,54,57], influenza-virus-induced pneumonitis [58], Streptococcus pneumoniae interaction with the respiratory tract [59], fibrotic lung disease, including cystic fibrosis [60,61], surfactant deficiencies [62], Hermansky–Pudlak syndrome type 2 [63], and small-cell lung cancer [64].

## 3. Idiopathic Pulmonary Fibrosis

IPF is a chronic and progressive fibrotic lung condition and the most common of idiopathic interstitial pneumonia. In Europe and North America, the incidence is estimated to range between 2.8 and 18 cases per 100,000 people per year [65,66]. IPF is more common in men and is rare in people younger than 50 years. The median survival time after diagnosis is 2–4 years. IPF is a consequence of multiple genetic and environmental risk factors leading to repetitive cycles consisting of local micro-injuries (damage) and aberrant regeneration of the alveolar epithelium. Because of altered epithelial–mesenchymal communication, continuous production of components of the extracellular matrix by myofibroblasts leads to aberrant fibrotic remodeling of the lung interstitium in detriment of functional respiratory tissue. This heterogeneous cellular context makes it very difficult to replicate through PSCs-derived minilungs. Chen et al. 2017, however, reported modeling of Hermansky–Pudlak syndrome (HPS) from PSCs-derived minilungs [53]. HPS is an autosomal recessive hereditary disease that may be complicated by progressive and potentially fatal interstitial pneumonia, characterized by an inflammatory process within the interstitial walls rather than the alveolar spaces [67]. HPS causes HPS-associated interstitial pneumonia (HPSIP), which is similar to IPF. HSP1 mutant organoids, generated by CRISPR/Cas9 technology, exhibited increased accumulation of mesenchymal content and enhanced deposition of collagens and fibronectin. This model recapitulated relatively well the clinical features present in HPSIP patients. Later, in 2019, Strikoudis et al. managed to introduce more HSP mutations associated with HPSIP into bona fine ESCs lines to create an IPF-like model of fibrogenic lung disease [61]. The mutant minilungs exhibited upregulation of interleukin-11 (IL-11) in ATII cells predominantly [61]. Athough the formal modeling of IPF is still elusive, this work is a good example of the applications of PSCs and organoid biotechnology to model complex respiratory and fibrogenic contexts. The fact that many of the IPF-associated mutations occur in genes encoding surfactant proteins (SFTPC or SFPTA2) has suggested direct involvement of ATII in pathogenesis of IPF [68,69,70,71]. In fact, the specific injury to ATII cells is a preeminent fact of the central fibrotic hypothesis regarding IPF [31,69,72,73,74]. Thus, damage inflicted on ATII cells can cause their entry into a state called cell senescence, mainly characterized by absence of proliferation [75,76,77]. The complex spectrum of molecules secreted by senescent cells leads to an exacerbated pro-inflammatory response, recruitment of inmmunitary cells, and fibrogenic activation of fibroblasts. Unfortunately, in the context of IPF, persistent damage and tissue repair lead to continuous fibrotic remodeling of the functional respiratory tissue and its aberrant regeneration. In addition, telomeric damage is a very important factor in development of IPF since 8–15% of patients with familial IPF have heterozygous mutations in genes associated with telomere maintenance and integrity, such as hTERT (reverse transcriptase) or hTERC (RNA component) [78,79,80]. Thus, the association between telomeric damage in ATII cells also contributes to alter their self-renewal and replacement of damaged ATI cells. ATII cells can enter easily a senescent state after genotoxic insults, such as that provoked by antibiotic bleomycin, which induces very efficiently DNA double-strand breaks (DSBs) [81,82,83], making bleomycin a very interesting model to analyze DNA damage. We and others have established in vitro and in vivo models of pulmonary fibrosis based on bleomycin, including ATII cellular systems, myofibroblasts, and mice sensitive to bleomycin-induced lung injury [83,84,85,86]. It should be noted here that, although bleomycin reproduces well many aspects of general pulmonary fibrosis and some lesions present in IPF, it has never been promoted as an experimental equivalent of IPF. However, the potential of bleomycin in induction of DSBs and senescence in many cell types is extraordinary. DSBs are excellent inducers of cellular senescence and can be accurately measured by different biotechniques. Expression of DNA damage and cellular senescence represent one of the early milestones of fibrogenic conditions, such as IPF, and many other conditions that evolve towards fibrosis [76,77,87]. We made use of the advantages of bleomycin to analyze expression of DNA damage in minilungs derived from hESCs. Moreover, we explored the influence of various vitamin-D-less hypercalcemic analogs that maintain their antifibrotic properties and act as very efficient DNA damage erasers [88]. This work represented a good example of use of PSC-derived minilungs to model early events that appear in the context of IPF and other fibrogenic conditions and as a platform for screening compounds of interest. Wilkinson et al. presented a method for generation of self-assembled human lung tissue and its potential for disease modeling and drug discovery [89]. They mounted a cohesive organoid from collagen-functionalized alginate beads and human fibroblasts in a rotational bioreactor, leading to structures recapitulating the native lung. Treatment with TGF-β1 showed a progressive scarring phenotype that resembles IPF. Schruf et al. generated a hiPSC-derived 2D airway-liquid interphase (ALI) culture model of ATII cell differentiation exposed to a pro-fibrotic environment to recapitulate phenotypic and functional features of aberrant epithelial remodeling in IPF lesions. The pro-fibrotic cocktail used was based on upregulated cytokines found in IPF patient bronchoalveolar lavage or sputum [90]. Another model for pulmonary fibrosis was reported by Suezawa et al., consisting of a co-culture model named fibroblast-dependent alveolar fibroblasts (FD-AOs) from hPSCs and primary human fetal lung fibroblasts. Recapitulation of epithelial–mesenchymal interactions by bleomycin treatment will serve for screening therapeutic agents to treat IPF [91].

## 4. Cystic Fibrosis

CF is a multi-organ genetic disorder that affects more than 70,000 people worldwide. Although CF incidence varies by country, it has been estimated in 1/3000 births in Caucasians in North America and Europe (ECFS patient registry. 2022—https://www.ecfs.eu/sites/default/files/ECFSPR_Report_2020_v1.0%20%2807Jun2022%29_website.pdf, accessed on 20 February 2023; [92,93]). Mortality and morbidity in CF are mainly associated with lung dysfunction due to tissue rearrangements and fibrosis, recurrent infections, and inflammation. Development of small molecules to improve CFTR protein function, termed CFTR modulators, has substantially benefitted people with cystic fibrosis. Today, the median life expectancy of a CF patient is around 53 years [92,93,94] (ECFS patient registry. 2022; [95]). CF is caused by mutations in the gene encoding the cystic fibrosis transmembrane conductance regulator (CFTR), a chloride and bicarbonate ion transport channel. Deletion of a phenylalanine at position 508 (Phe508del or ΔF508) is the most common mutation and represents more than two-thirds of all mutations [96,97,98,99,100,101,102]. In the lung, CFTR dysfunction deregulates transport of chloride and bicarbonate into the lumen of the airways, which normally regulates reabsorption of Na^+^ mediated by another channel (Na^+^ channel, or ENac). This leads to net water uptake by the respiratory epithelium, which results in dehydration of the liquid surface of the airways [103,104]. Consequently, the mucus of the mucociliary ladder becomes dehydrated and its clearance is compromised. A second hypothesis on CFTR dysfunction is related to the pH of the airway surface liquid. CFTR dysfunction would reduce bicarbonate secretion into the airway lumen, resulting in a decreased pH [105,106,107]. In any case, the CF lung context provides an excellent niche for colonization and infection of opportunistic pathogens and compromised antimicrobial defense. Thus, chronic airway obstruction, infection, and inflammation account for the majority of morbi-mortality associated to CF [108,109]. Regarding fibrosis, tissue remodeling with increased collagen deposition is common in distal airways of CF patients [110,111].

Modeling of CF through hPSCs also represents a challenge as CF affects specific cell subtypes in different regions of the lung. MacCauley et al. reported in 2017 generation of airway organoids from hPSCs based on specific modulation of the WNT pathway to achieve specification of respiratory progenitors [60]. The WNT pathway is involved in regulation of the proximodistal pattern of the human airways. In this remarkable work, the authors generated a new “low-Wnt” distal organoid differentiation protocol from hiPSCs-derived NKX2.1^+^ lung progenitors. They also applied CRISPR and TALENS gene editing on hiPSCs of genotype ΔF508/ΔF508, with a defect in forskolin-induced swelling, to correct the disease mutation. This approach engaged many applications in modeling and drug screening for airway diseases, such as CF and primary ciliary dyskinesia. Later, in 2020, Geurst et al. showed generation of a CF’s patient-derived intestinal organoid biobank to study gene editing mediated by adenine base editors (ABE editing). Conventional CRISPR/Cas9-mediated genome editing depends on introduction of DNA double-strand breaks (DSBs) at the target site [112]. The novel technology presented, based on versions of Cas9 endonucleases (SpCas9-ABE and xCas9-ABE), enables enzymatic conversion from A–T into G–C base pairs without introducing DSBs. This novel CRISPR-Cas9 technology enables efficient correction of mutations without genome-wide off-target effects and represents promise for hereditary diseases [112].

## 5. Chronic Obstructive Pulmonary Disease

COPD is characterized by airflow limitation and abnormal inflammatory response of the airways to noxious particles and gases [113]. Currently, COPD is the third leading cause of death only behind ischemic heart disease and stroke (World Health Organization). The main causes of COPD include smoking tobacco, lung growth, environmental stimuli, and a complex background of genetic factors. The progressive course of COPD is frequently aggravated by exacerbations that reduce quality of life, accelerate disease progression, and increase risk of death. Several causes of exacerbations have been suggested, such as heart failure, pneumonia, pulmonary embolism, some medications, or inhalation of irritants. The most frequent cause of exacerbation is viral or bacterial infection [113,114].

The principal pathophysiological feature of COPD is obstruction of the airways. This is caused by increased mucus content, mucosal hyperplasia, infiltrations of inflammatory cells, and fibrotic remodeling due to excessive connective tissue deposition in the peribronchial space [18,115,116]. This progressive obliteration of the respiratory bronchioles is eventually accompanied by emphysema, which typically starts in the bronchioles. Thus, pulmonary emphysema and fibrosis are combined events that were first characterized in a homogeneous group of patients with both emphysema and interstitial lung disease (ILD) with pulmonary fibrosis in the lower lobes. Moreover, pulmonary emphysema is related to cellular senescence in terms of proliferation arrest, increased inflammation due to the pro-inflammatory properties of the senescent-associated secretory phenotype (SASP), aberrant cell regeneration, and fibrotic remodeling, and, eventually, carcinogenesis. Cigarette smoke and oxidative stress are also inducers of cell senescence; thus, COPD can be interpreted as a condition that accelerates aging, with several aging pathways involved in its pathogenesis [17,18,117,118,119]. Cigarette smoke can induce expression of senescence marker p21 in epithelial cells and fibroblasts. In addition, lungs with emphysema also show increased expression of p16, p19, and p21, all of which are cyclin kinase inhibitors and markers of cell senescence. Moreover, cigarette smoke exposure and mitochondrial dysfunction have been shown to lead to oxidative stress in COPD [120]. Oxidative stress arises primarily from presence of radical oxygen species (ROS). ROS are excellent inducers of DNA damage, DSBs being the most deleterious. Expression of DNA damage in lung cells and induction of permanent DNA damage response represent other hallmarks of the cellular senescence phenotype contributing to losing regenerative capacity and repair and progressive worsening of lung function [121,122].

CF and COPD share various characteristics, such as progressive airflow obstruction, chronic airway inflammation in the lumen, and recurrent infectious exacerbations, suggesting the possibility of common mechanisms [123]. Although COPD is mainly caused by environmental factors on a genetically susceptible background, the ideal modeling from PSCs and derived organoids would require similar strategies to those reported by MacCauley et al. for CF modeling [60]. Several works have reported generation of iPSCs in the context of COPD [124,125,126]. Ahmed et al. 2022 [124] managed to generate hiPSCS from peripheral blood mononuclear cells and differentiated them into an ALI bronchial epithelium without using purification of airway progenitors by cell sorting. This epithelium showed large zones with beating ciliated, basal, goblets, club cells, and neuroendocrine cells. However, we are still witnessing an initial development of adequate models for faithful modeling of the events that define COPD. These events include, among others, mucin production as observed in smokers and COPD patients, events related to fibrotic remodeling, and cellular senescence. This work represents an ideal experimental platform that could be enriched with mesenchymal cells, i.e., myofibroblasts, or used to assess the effect of tobacco smoke and pollution particles. Moreover, polymicrobial infections would help to reproduce the major exacerbations occurring in COPD patients. Figure 2 illustrates the usefulness of minilungs derived from PSCs to model COPD with exacerbations and other diseases.

## 6. Severe Acute Respiratory Syndrome Coronavirus Type 2 (SARS-CoV-2) and Fibrosis

Another interesting context that can potentially be modeled and exploited via PSCs-derived minilungs is the one represented by severe acute respiratory syndrome coronavirus type 2 (SARS-CoV-2). COVID-19 patients may be asymptomatic or show symptoms such as those agglutinated in acute respiratory distress syndrome (ARDS) and pulmonary fibrosis. Pulmonary fibrosis is now recognized as a sequel of ARDS. Fibrotic lesions can be observed in high-resolution chest tomography (CT) scans of patients recovered from COVID-19, including ground glass opacity or a combination of irregular interlobular septal thickening and mild traction bronchiectasis [127,128,129,130,131,132]. These imaging findings are supported by autopsy reports. Reports on patients who died of COVID-19 pneumonia reveal features of diffuse alveolar damage with areas of consolidation by fibroblastic proliferation and deposition of extracellular matrix and fibrin in alveolar spaces [130]. We also know that patients with respiratory diseases compared to those without respiratory diseases have higher risk of hospitalization due to SARS-CoV-2 infection [133]. Moreover, patients with fibrotic interstitial lung disease (fibrotic ILD), especially those with IPF, have a higher risk of death after infection with SARS-CoV-2 [134]. All ILDs patients, especially IPF patients, can undergo acute exacerbations of different origins: internal accelerations of the fibrotic conditions, external events leading to acute lung injury, and diffuse alveolar injury. Acute lung damage can be triggered by viral infections and greatly enhanced by immune responses of the host. Respiratory failure in IPF patients due to these exacerbations accounts for higher hospital mortality, estimated in more than 50% in most cases [135,136]. The potential molecular mechanisms associated with the replicative cycle of SARS-CoV-2 by which fibrosis develops include [137]:(i)Interaction of viral spike “S” protein with angiotensin converting enzyme (ACE2) receptor. Binding of the virus to its receptor can downregulate level of ACE2, increase levels of Ang II, and decrease level of Ang1–7, thus promoting inflammation and fibrosis.(ii)Aberrant immune response leading to so-called “cytokine storm”, resulting in increased plasma levels of IL-1β, IL-2, IL-7, and IL-10, GCSF, MIP1α, IFNy, IP-10, IL-6, IL-8, TNFα, etc. Cytokine storm accelerates disease progression and aggravates ARDS and multiple organ failure. Release of pro-inflammatory cytokines and metalloproteinases during ARDS induces damage of the epithelium, endothelium, and fibrotic remodeling.(iii)Infection and damage of ATII cells. ATII cells are key mediators of the alveolar innate response and regeneration of the respiratory epithelia through its proliferation and differentiation into ATI cells. ATII can enter the senescence state and secrete a series of inflammatory mediators and metalloproteinases (SARS phenotype) that mediate tissue remodeling. For instance, TGFβ triggers proliferation and differentiation of fibroblasts into myofibroblasts, causing aberrant deposition of extracellular matrix proteins during abnormal fibrosis.(iv)Damage of endothelial cells. Endothelial cells, when injured, transform into a mesenchymal state (EndMT) with increased activity of mesenchymal protein secretion and matrix metalloproteinases, leading to accumulation of fibroblasts and myofibroblasts and induction of fibrotic remodeling in the interstitium of the lung. Transit of the virus into the lower respiratory tract the lung can be favored using mechanical ventilation. Presence of the virus in the lung may contribute to acute lung injury and secondary fibrosis.

The course of the COVID-19 pandemic accelerated the search for suitable study models based on hPSCs. Yang et al., 2020 [138] proposed an experimental platform comprising cells and organoid derivatives of hPSCs to study SARS-CoV-2 tropism and infectivity. They studied permissiveness to viral infection of pancreatic endocrine cells, liver organoids, cardiomyocytes, and dopaminergic neurons, all derived from hPSCs. However, their platform did not include any airway or lung organoids, greatly limiting its impact on understanding viral interactions with the respiratory tract and the study of fibrotic events. In 2021, Han et al. [139], however, developed human minilungs and colonic organoids from hESCs (RUES2) based on previously reported stepwise strategies. As expected, both the minilungs and the colonic organoids generated were permissive to infection of SARS-CoV-2. A remarkable aspect of this work is use of this platform for high-throughput drug screening to identify candidate COVID-19 therapeutics. The authors identified several drugs that inhibit SARS-CoV-2 entry, including imatinib, MPA, and QNHC, both in vitro and in vivo.

Bidimensional arrays of the airways and lung in different formats (multi-well plates, inserts, with or without an ALI) easily enable enrichment with other cell types, such as fibroblasts, myofibroblasts, and endothelial cells. These co-cultures, more or less complex, enable the study of the molecular events associated with infection in COVID-19 and elucidation of mechanisms underlying expression of DNA damage, cell senescence, and lung fibrosis. Generation of 2D minilungs enriched with myofibroblasts allowed us to study the infectivity of both epithelial cells and myofibroblasts by human respiratory syncytial virus (HRSV) [54]. Recently, we reported the usefulness of 2D and 3D minilungs to study the influence of DNA damage after treatment of potential pro-fibrotic compounds, such as vitamin D3, and some less hyper-calcemic vitamin D analogs [88]. Schruf et al. [90] managed to develop a more sophisticated co-culture model or “EpiAlveolar co-culture model” consisting of human primary alveolar epithelial cells, fibroblasts, and endothelial cells, with or without macrophages, to serve as a platform to predict long-term responses to aerosols. This work is a good example of how histological alterations can be modeled in hiPSC-derived ATII-like cell cultures maintained in ALI as a suitable model to study fibrosis. Proinflammatory and profibrotic responses could be assessed upon repeated subchronic exposures to different compounds. Other interesting experimental approaches to study molecular events in COVID-19 are based on use of Lung-On-ChiP (LOC) technologies consisting of biomimetic microsystems that reconstitute the critical functional alveolar–capillary interface of the human lung. These co-cultures are suitable to study the sequential events in COVID-19, at the air–blood barrier, in the presence of alveolar cells and peripheral immune cells and other cell types of interest [140].

## 7. Discussion

Respiratory diseases represent a health problem of the first order and are one of the leading causes of global morbidity and mortality. Currently, there is no cure for most respiratory diseases, which are treated symptomatically. Thus, there is a critical need for adequate study models to delve into knowledge of the disease and explore novel therapeutic strategies. Research of human lung disease relied on cultures of immortalized cell cultures and animal models that, in many cases, did not faithfully reflect the human context. Isolation of primary cells from fragments of human material from the airways or the lung introduced new cell systems. These cultures reproduced relatively well the cellular context of the airways and lung displaying cells in bidimensional arrays or 3D structures, such as tracheospheres, bronchospheres, alveolospheres, etc., growing in suspensions or embedded in suitable hydrogels. The main drawbacks of these systems were availability of precious human material and the cell spectrum limited to a few types. The advent of PSCs biotechnology marked a qualitative leap in the field of airways and lung organoids and has increased our knowledge on the pathophysiology of the human lung and expanded our capacity and prospects for disease modeling and regenerative therapy.

Exploitation of the cellular pathways that control lung development has enabled generation of airways and lung organoids through sequential differentiation of PSCs. The stepwise protocols available rely on generation of definitive endoderm and further differentiation to anterior foregut endoderm (AFE) and ventral AFE. Lung and airway progenitors can be further differentiated into mature epithelial cell types using FGFs, glucocorticoid agonists, etc. Essentially, the signaling pathways involved are those implicating TGFβ, WNT, BMP4, Activin A, Retinoic acid, glucocorticoids, and FGFs [46,47,48,49,50,51,52,53,56,60,62,141,142,143,144]. A plethora of signaling pathways have been described in animal models [141,142,145,146,147] and exhibit high levels of temporal and regional specificity by which they each promote differentiation and maturation of specific cell types at the expense of others.

Modeling of respiratory diseases that evolve with some type of fibrosis of different magnitude and location is more challenging as it implies complex interactions between epithelial and mesenchymal cells. Differentiation of PSCs into lung and airways cells is normally accompanied by mesodermal cells. The source of mesoderm is probably due to a small population of contaminant cells that merged from spurious differentiation. It is well known that, during in vitro expansion of PSCs, colonies (“bad ones”) may be generated, with suboptimal quality. In addition, the differentiation efficiency of PSCs to any of the germ layers is rather far from 100%. Other possible sources of mesenchymal cells could be presence of traces of early mesendoderm specification in the definitive endoderm generated. Several reports have shown that the differentiation procedure will inevitably lead to presence of co-derived non-lung lineages [43,46,48,49,51,52,56,144]. Three noteworthy works are those reported by McCauley et al., Chen et al., and Stridoukis et al. [53,60,61]. The approach described by McCauley et al. produced reliable production of “epithelial-only” clean human airway organoids from hiPSCs to model CF and other genetic disorders [60]. Modeling of HPSIP, however, a clinical IPF-like disease described first by Chen et al. and then followed by Stridoukis et al., is based on appearance of mesodermal content in their HPS-mutant minilungs [53,61]. These mutant organoids expressed markers of extracellular cell matrix (ECM) and mesenchymal markers and a particular gene expression signature resembling the one of human IPF and rat lungs treated with bleomycin.

## 8. Future Directions

Currently, there is a need to find suitable models of respiratory diseases, especially those that evolve with more or less fibrosis. This is a challenge as modeling of fibrotic conditions implies replication of complex epithelial and mesenchymal cells interactions. PSCs-based models are already a reality and enable generation of cells containing phenotypic and functional markers or mature airways and lung epithelial cell types. The derived organoids will help to better understand the disease, reduce use of animal models, and serve as a platform for drug screening and regenerative medicine.

Among the main advantages of these models are unlimited availability of differentiated material, the possibility of gradually increasing the cellular spectrum by specific enrichment with cell types of interest, and the possibility of making individual-specific organoids. These specific contexts will enable application of gene editing strategies to correct mutations prior to production of organoids for disease modeling, drug screening, or subsequent in vivo tissue regeneration. However, PSCs differentiation also has several potential disadvantages, such as limitless differentiation capabilities that are difficult to control and their inherent ability to develop spurious differentiation. Differentiation protocols and large-scale production of hPSCs will need to be optimized and investigated prior to clinical use. Use of hPSCs carries ethical concerns, and both ESCs and iPSCs have been shown to form teratomas [148,149]. Nonetheless, hPSCs can be used to generate specific lung progenitors so that gene editing techniques—large-scale production from bioreactors and tissue bioprinting, for instance—could be applied and implemented with more efficiency. It has been reported that iPSCs partially retain the epigenetic memory of their tissue of origin, which could lead to limitations [150].

The large variety of organoid formats available, such as bidimensional arrays, ALI cultures, and enriched co-cultures on inserts, or even LOC biotechniques, on which high-throughput techniques can be applied, will enable rapid analysis of a multitude of biological measurements. These techniques will enable accurate analysis of the infectivity features of pathogens in specific genetic contexts, tissue damage, DNA damage and oxidative stress expression, cell senescence, and pro-fibrotic and pro-inflammatory responses.

## Figures and Tables

**Figure 1 ijms-24-04413-f001:**
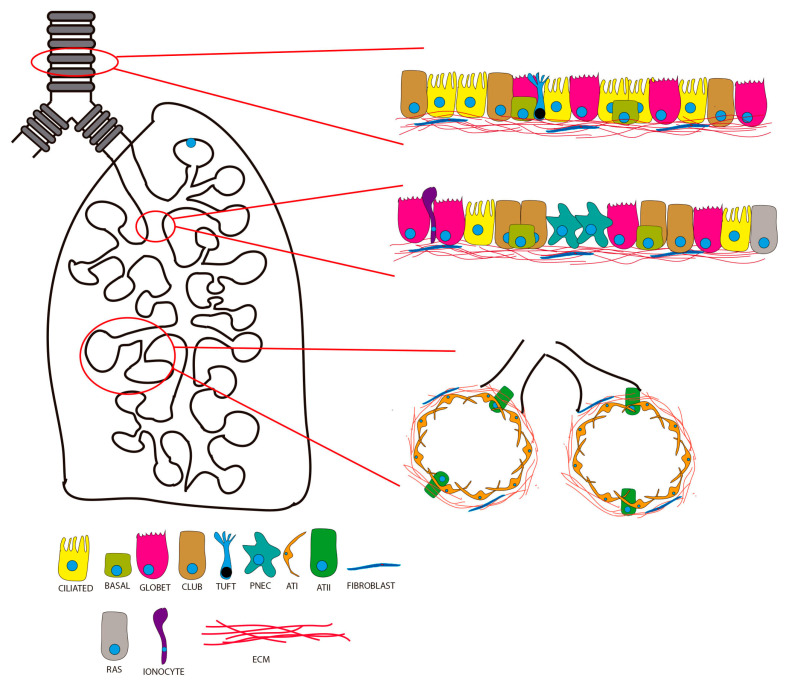
Schematic representation of the different compartments of the respiratory system. The panels on the right show the cell composition of the trachea and upper airways (upper panel), the upper bronchioles (middle panel), and the alveoli (bottom panel). PNEC: pulmonary neuroendocrine cell; ATI: alveolar type I cell; ATII: alveolar type II cell; RAS: respiratory airway secretory cell; ECM: extracellular matrix.

**Figure 2 ijms-24-04413-f002:**
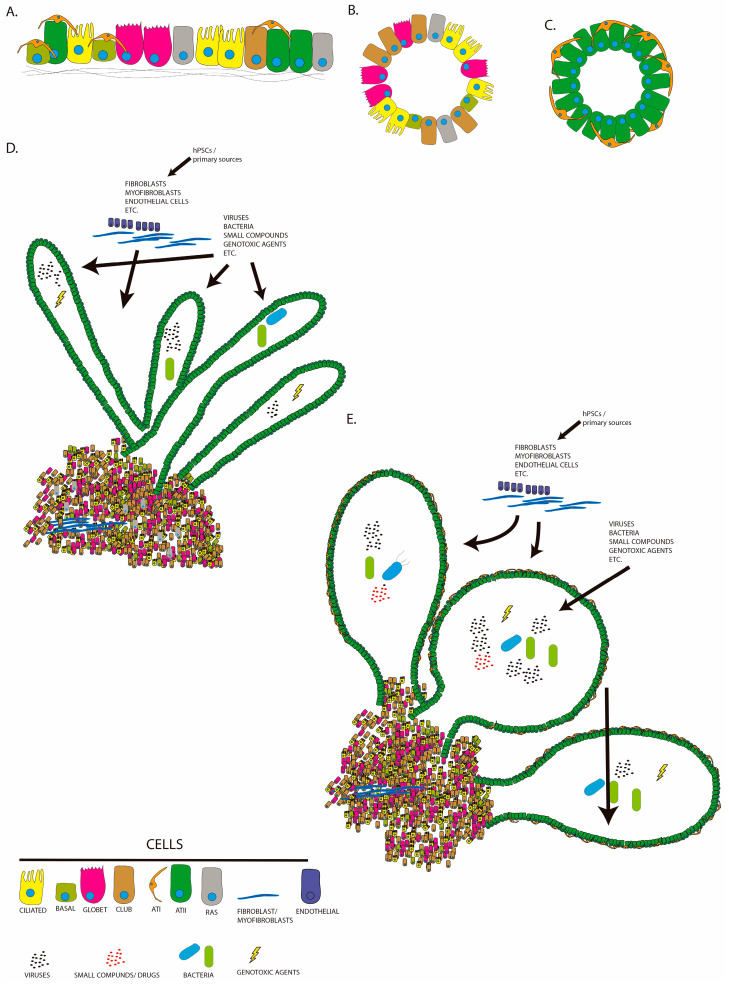
Illustrations of some examples of respiratory organoids that can be derived from PSCs. (**A**) Bidimensional arrays (2D minilungs) of both airways and lung epithelial cells derived from PSCs mounted on different substrates, such as human fibronectin. Tridimensional structures composed of cells of the airways (**B**) or the alveoli (**C**), derived from PSCs and embedded into biogels (e.g., Matrigel^Tm^, etc.). “Lung buds” organoids [53] (**D**) or paddle-racquet lung organoids (PRLOs [55]) (**E**) composed of both airways and lung epithelial cells derived from PSCs. Proximal areas are mainly composed of cells from the airways with minor mesenchymal content. Distal areas represent the alveoli and are enriched in ATII cells, which may eventually undergo differentiation to ATI cells. These minilungs can be enriched with additional cell types, such as endothelial cells or fibroblasts and/or myofibroblasts from primary sources or the differentiation of PSCs. These minilungs can be used to model any respiratory disease or perform drug screening. Infection agents or genotoxic agents, such as bleomycin or liquid cigarette smoke, can be delivered into the lumen of the organoids by means of microinjections. RAS: respiratory airway secretory cells.

## Data Availability

Not applicable.

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
