# Peer review of "Modeling of Respiratory Diseases Evolving with Fibrosis from Organoids Derived from Human Pluripotent Stem Cells"

_ijms, 2023, doi:10.3390/ijms24054413_

Round 1
Reviewer 1 Report
The topic is very interesting, actual and complex. Pulmonary fibrosis is a manifestation of various diseases. The major point I miss in the review is a presentation of the pathology mechanisms and its cellular actors; this has also an influence on the development of the in vitro models. The manuscript would benefit from table/figures etc. It certainly worth to rethink about the best possible way to present the manuscript.

Author Response
POINT-BY-POINT RESPONSES TO THE REVIEWERS
Dear editors,
We would like to thank you and the reviewers for the editorial management, comments and suggestions on the paper. Below please find our point-by-point response to the reviewers.
Sincerely
Alberto Zambrano
Reviewers’ comments:
Reviewer #1 (Comments for the Author)
The topic is very interesting, actual and complex. Pulmonary fibrosis is a manifestation of various diseases. The major point I miss in the review is a presentation of the pathology mechanisms and its cellular actors; this has also an influence on the development of the in vitro models. The manuscript would benefit from table/figures etc. It certainly worth to rethink about the best possible way to present the manuscript.
The abstract is very general:
“respiratory disease” should be plural since there is not a single respiratory disease
“These novel human pluripotent stem cell-derived organoids have enabled relatively
accurate disease modeling. In many respiratory conditions, such as those evolving with
more or less airways or lung fibrosis, the disease modeling is a real challenge due to the large number of epithelial cells involved and the interaction with other cell types of
mesenchymal origin. I would advice to focus immediately on the diseases you are going to further talk about in your review.
Our response: The abstract has been modified as suggested.
If The focus is modelling fibrosis: an overview should be given of the pathological process and the cellular component involved. Which components are important and should be present also ideally in the mini-organ-lungs?
In the introduction a very general overview of the normal anatomy is given but no more than this. The pathological mechanism is completely ignored. Later on, pathological components are introduced but not explained.
Our response: An overview about fibrosis has been included.
Introduction
Line 25 to 45: this is very basic knowledge: it would be nice to have a figure representing the repiratory system, depicting also the new knowledge and to introduce references.
Our response: We have included two Figures. Figure 1 illustrates the respiratory system and its cellular composition.
Line 42-43: 2 important papers are cited but very little is said about the exciting finding
And it is not reported the location of theses cells and also why these finding s are important
Our response: The relevance of these new cells are mentioned.
Line 45 a reference is needed
Our response: two references have been included.
Line 69-73: the question, aim of the review is not reflect in the abstract. I suggest to focus the abstract to the aim of your review. I do not find anywhere how the review was conducted, the method for this review and
selection of the presented literature
Our response: The abstract has been modified as suggested. Aim and diseases included
Line 49-51: “Causes of pulmonary fibrosis include environmental pollutants, some drugs, connective tissue diseases, or interstitial lung disease, however, in most cases, no clear cause is found” There are no references
Our response: We have added references.
Line 51-59: no references. They are needed.
Our response: We have added references.
Paragraph starting from line 61: iPSCs embryonic and adult stem cells are mentioned but it is not mentioned how/why they hare chosen for modelling.
Line 65: no references
Our response: the advantages of PSCs over ASCs are mentioned. Reference added.
From line 69: specific diseases are mentioned later on but in this paragraph the focus is not presented
Our response: the representative diseases, focus of the review, are included.
Line 76-77: it is a repetion, aalready said in few lines above in the introduction
Our response: we have remove those sentences.
Line 103: 2D appears for the first time but no indication is given about use of 2D or 3D
Our response: the 2D arrays are now mentioned first at the beginning of the section (“…Wnt agonist CHI99021 and the BMP4 inhibitor Noggin are commonly used to generate lung progenitors that are subsequently matured by the addition fibroblast growth factors FGF2, FGF7 and FGF10 and BMP4 [9, 12, 14-18]. Thus, PSCs are driven to produce epithelial cells from the airways and alveoli. The differentiated cells can be displayed as a bidimensional array (2D minilung) or in 3D.”
I miss somewhere why it is important to invest in these invitro models since still currently lot s of experiments are done on laboratory animals: this part would add an extra dimension to your review. The description of the in vitro model would benefit of an explanatory figure.
Our response: we have added two Figures. Figure 2 illustrates some in vitro structures.
Line 118: myofibroblasts appear for the first time here but no information are given about location and importance; are they part of normal anatomy or this is already pathology?is the damaged alveolar epithelium replaced by myofivroblast, I would not know how to deal with it here
Our response: An overview about fibrosis has been included in the introduction.
Line 121: HPS is now presented, but what is it exactly? Also the terminology “interstitial pneumonia” is not explained
Our response: HPS and interstitial pneumonia is now mentioned.
Line 128: which kind of mutations?
Our response: mutations are described in the reference included.
“Later, in 2019, Strikoudis et al. managed to introduce more HSP mutations associated to HPSIP into bona fine ESCs lines to create an IPF-like model of fibrogenic lung disease [61]. The mutant minilungs exhibited the upregulation of interleukin-11 (IL-11) in ATII cells, predominantly [61].”
Line 151: reference 49 and references therein. Does it mean that the reader has to consider all references in this paper? Please be more precise. What does it mean susceptible mice? We and others: but only a refence is mentioned
Our response: We have selected some references from the article.
There a reference needed in line 182.
Our response: References have been included.
End of line 219; references are nee
Our response: Reference included.

Reviewer 2 Report
In this review, Irene Chamorro-Herrero and Alberto Zambrano provide a comprehensive, balanced, and high-quality summary of the state of the field of human pluripotent stem cell (PSC)-derived lung organoids. They describe the use of these organoids to model major lung diseases, including interstitial lung disease. Overall, the document is well written and well documented. A weak point is that sometimes this review is more of a general review on respiratory diseases than a review focused on lung organoids, as for example in chapter '6. Severe acute respiratory syndrome coronavirus type 2 (SARS-CoV-2) and fibrosis”, but this does not call into question the overall quality of the work.
I have a few minor remarks, listed below:
1/ Replace the term “Clara cells” with “Club cells” (https://pubmed.ncbi.nlm.nih.gov/23321606)
2/ Line 36: in humans, club cells are not the dominant cell type, even in the small airways, unlike mice (https://pubmed.ncbi.nlm.nih.gov/30930931). Please rephrase.
3/ I would not define senescence by "persistent response to DNA damage" (line 136). Although DNA damage is indeed a cause of senescence, it is not the only one. Please rephrase.
4/ Line 184: “Today, the median life expectancy for a patient with cystic fibrosis is approximately 30 years [55–57]. This is no longer true with CFTR modulators, life expectancy has increased dramatically. Please correct and update references.
5/ Line 317: “The transit of the virus into the lower respiratory tract the lung is favored by use of mechanical”. Correct the sentence.
6/ Line 322: study permissiveness: put in the past.
7/ Permissiveness or permissiveness?
8/ Line 365: “These systems could reproduced relatively well the pulmonary”. Please correct.
9/ Line 418: ‘limited differentiation efficiency’: I disagree with this idea. PSCs have very efficient and broad differentiation abilities. Rather, the problem is the limitless differentiation capabilities of the PSC, which makes it more difficult to drive. Please rephrase.
10/ Line 424: ‘It has been reported that iPSCs they partially retain the epigenetic memory of their tissue of origin that could lead to limitations’. Please delete "they".
Author Response
POINT-BY-POINT RESPONSES TO THE REVIEWERS
Dear editors,
We would like to thank you and the reviewers for the editorial management, comments and suggestions on the paper. Below please find our point-by-point response to the reviewers.
Sincerely
Alberto Zambrano
Reviewers’ comments:
Reviewer #2 (Comments for the Author)
Reviewer #2
In this review, Irene Chamorro-Herrero and Alberto Zambrano provide a comprehensive, balanced, and high-quality summary of the state of the field of human pluripotent stem cell (PSC)-derived lung organoids. They describe the use of these organoids to model major lung diseases, including interstitial lung disease. Overall, the document is well written and well documented. A weak point is that sometimes this review is more of a general review on respiratory diseases than a review focused on lung organoids, as for example in chapter '6. Severe acute respiratory syndrome coronavirus type 2 (SARS-CoV-2) and fibrosis”, but this does not call into question the overall quality of the work.
I have a few minor remarks, listed below:
1/ Replace the term “Clara cells” with “Club cells” (https://pubmed.ncbi.nlm.nih.gov/23321606)
Our response: “Clara cells” have been changed to “Club cells”.
2/ Line 36: in humans, club cells are not the dominant cell type, even in the small airways, unlike mice (https://pubmed.ncbi.nlm.nih.gov/30930931). Please rephrase.
Our response: We have changed the sentence: “Club cells are secretory cells of the bronchiolar epithelium and contribute to the production of non-mucinous to the extracellular lining fluid”
3/ I would not define senescence by "persistent response to DNA damage" (line 136). Although DNA damage is indeed a cause of senescence, it is not the only one. Please rephrase.
Our response: We have modified the sentence:
Thus, damage inflicted on ATII cells can cause their entry into a state called cell senescence, mainly characterized by the absence of proliferation (ref).
4/ Line 184: “Today, the median life expectancy for a patient with cystic fibrosis is approximately 30 years [55–57]. This is no longer true with CFTR modulators, life expectancy has increased dramatically. Please correct and update references.
Our response: It has been modified accordingly.
5/ Line 317: “The transit of the virus into the lower respiratory tract the lung is favored by use of mechanical”. Correct the sentence.
- Our response: We have modified the sentence. “The transit of the virus into the lower respiratory tract the lung can be favored by the use of mechanical ventilation. The presence of the virus in the lung may acute lung injury and secondary fibrosis”.
”
6/ Line 322: study permissiveness: put in the past.
Our response: Sentence changed accordingly.
7/ Permissiveness or permissiveness?
Our response: permissiveness
8/ Line 365: “These systems could reproduced relatively well the pulmonary”. Please correct.
Our response: We have corrected the sentence: “These cultures reproduced relatively well the cellular context of the airways and lung displaying cells in bidimensional arrays or in 3D structures such as tracheospheres, bronchospheres, alveolospheres, etc., growing in suspensions or embedded into suitable hydrogels”..
9/ Line 418: ‘limited differentiation efficiency’: I disagree with this idea. PSCs have very efficient and broad differentiation abilities. Rather, the problem is the limitless differentiation capabilities of the PSC, which makes it more difficult to drive. Please rephrase.
Our response: The sentence has been modified as suggested. “However, PSCs differentiation also have several potential disadvantage such as the limitless differentiation capabilities that are difficult to control and their inherent ability to develop spurious differentiation.”
10/ Line 424: ‘It has been reported that iPSCs they partially retain the epigenetic memory of their tissue of origin that could lead to limitations’. Please delete "they".
Our response: WE have deleted “they”
